

# Detection and benchmarking of somatic mutations in cancer genomes using RNA-seq data

Alexandre Coudray[1], Anna M. Battenhouse[2], Philipp Bucher[1] and Vishwanath R. Iyer[3]

[1] School of Life Sciences, École Polytechnique Federale de Lausanne, Lausanne, Switzerland
[2] Department of Molecular Biosciences, University of Texas at Austin, Austin, TX, USA
[3] Department of Molecular Biosciences, Center for Systems and Synthetic Biology, Institute for Cellular and Molecular Biology, Livestrong Cancer Institutes, University of Texas at Austin, Austin, TX, USA

Corresponding author
Vishwanath R. Iyer,
vishy@utexas.edu

## ABSTRACT

To detect functional somatic mutations in tumor samples, whole-exome sequencing (WES) is often used for its reliability and relative low cost. RNA-seq, while generally used to measure gene expression, can potentially also be used for identification of somatic mutations. However there has been little systematic evaluation of the utility of RNA-seq for identifying somatic mutations. Here, we develop and evaluate a pipeline for processing RNA-seq data from glioblastoma multiforme (GBM) tumors in order to identify somatic mutations. The pipeline entails the use of the STAR aligner 2-pass procedure jointly with MuTect2 from genome analysis toolkit (GATK) to detect somatic variants. Variants identified from RNA-seq data were evaluated by comparison against the COSMIC and dbSNP databases, and also compared to somatic variants identified by exome sequencing. We also estimated the putative functional impact of coding variants in the most frequently mutated genes in GBM. Interestingly, variants identified by RNA-seq alone showed better representation of GBM-related mutations cataloged by COSMIC. RNA-seq-only data substantially outperformed the ability of WES to reveal potentially new somatic mutations in known GBM-related pathways, and allowed us to build a high-quality set of somatic mutations common to exome and RNA-seq calls. Using RNA-seq data in parallel with WES data to detect somatic mutations in cancer genomes can thus broaden the scope of discoveries and lend additional support to somatic variants identified by exome sequencing alone.

## INTRODUCTION

Cancer is among the leading causes of death worldwide, with 8.7 million deaths in 2015 (*Global Burden of Disease Cancer Collaboration, 2017*). As a genetic disease, cancers are driven in part by the accumulation of somatic mutations, which incidentally, also offer targets for new precision therapies directed against tumor-causing mutations (*The Cancer Genome Atlas Research Network et al., 2013*; *Yu, O'Toole & Trent, 2015*).

Cancer cells typically accumulate somatic alterations that impact specific pathways implicated in cell growth, survival, angiogenesis, motility and other hallmarks of cancer (*Hanahan & Weinberg, 2011*). Advances in next-generation sequencing technologies have allowed increasingly fast, accurate and cost-efficient analysis of DNA and RNA samples, which has driven the identification of key cancer-driving mutations (*Raphael et al., 2014*). These findings are beginning to pave the way for new targeted therapies in many cancers, but significant challenges remain (*Paez et al., 2004*; *Taylor, Furnari & Cavenee, 2012*).

The actual cancer-driving mutations need to be differentiated from somatic passenger mutations caused by impaired DNA repair mechanisms, inherited or de novo germline mutations and neutral polymorphisms, and artefacts that can arise from sequencing errors, PCR or misalignment (*Berger et al., 2016*; *Sahni et al., 2013*; *Takiar et al., 2017*). Moreover, the complex structure of tumors increases the complexity of the analysis, as tumors are typically heterogeneous, containing normal cells as well as distinct clonal lineages of tumor cells (*Meacham & Morrison, 2013*). Somatic alterations typically range from substitution mutations and small insertions/deletions (indels) to chromosome rearrangements and copy number variations (CNVs) (*Rhee et al., 2017*).

To detect mutations in a tumor sample, whole exome sequencing (WES) has generally been favored over whole genome sequencing (WGS) for its relatively low cost, although dropping costs of WGS encourage its use for somatic mutation identification (*Alioto et al., 2015*; *Puente et al., 2011*). Whole-transcriptome (RNA-seq) data has typically been used to measure gene expression and identify transcript and splicing isoforms. Nevertheless, it is possible to identify genomic variants from RNA-seq (*Piskol, Ramaswami & Li, 2013*). Previous studies examining the use of RNA-seq for somatic mutation detection have focused on the characteristics of mutational changes seen in RNA-seq versus WES, but these studies have been limited with regard to cancer type, and there has been little systematic evaluation of the biological novelty and significance of tumor somatic variants detected by RNA-seq (*O'Brien et al., 2015*).

Here, we assessed the utility of RNA-seq for somatic mutation detection in glioblastoma multiforme (GBM), the most common and deadliest form of adult primary brain cancer. GBM shows a median overall survival of only 14–15 months (*Stupp et al., 2009*). Standard of care for GBM has not changed for many years, and emerging new targeted therapies (mostly targeting angiogenesis-related pathways) unfortunately encounter problems of drug resistance (*Stavrovskaya, Shushanov & Rybalkina, 2016*), making the discovery of new target genes of great importance. We focused on the use of the STAR aligner (*Dobin et al., 2013*) which is fast and is transcript-aware, and therefore has the potential to give additional information about mutations in cancer-activated transcripts that might be missing in WES, and MuTect2 from GATK (*Cibulskis et al., 2013*) which has been widely used for mutation identification. Our analysis showed that RNA-seq is able to detect novel, GBM-related somatic mutations and can thus complement exome and whole-genome sequencing in identifying somatic mutations in tumor genomes.

## MATERIALS AND METHODS

### Methods overview, sample preparation, data origin and databases used

We developed a new pipeline to detect somatic mutations in RNA-seq data, combining RNA-seq alignment using a STAR 2-pass procedure with somatic mutation detection using MuTect2 for variant calling (*Cibulskis et al., 2013*). Variants from RNA-seq and WES were compared, first, on a pair of RNA-seq/WES from a GBM tumor that had already been analyzed in our laboratory (*Hall et al., 2018*) and then on a set of nine pairs of RNA-seq and WES data from GBM tumors analyzed by the Cancer Genome Atlas (TCGA) (*Brennan et al., 2013*). We compared and evaluated RNA-seq and WES mutations in four steps. First, we estimated the proportion of germline or somatic mutations by comparison of identified variants to the dbSNP database (*Kitts et al., 2013*) which catalogs known germline variants, and the Catalogue Of Somatic Mutations in Cancer or COSMIC database (*Forbes et al., 2015*), respectively. The use of these databases allowed us to evaluate whether a variant was a germline (included in dbSNP but not in COSMIC) or a somatic mutation (included in COSMIC but not in dbSNP). Second, somatic mutations detected in RNA-seq-only data were consolidated to highlight mutations present in multiple tumor samples. Third, their functional impact on proteins was evaluated by using two scoring systems: SIFT and functional analysis through hidden markov models (FATHMM) with cancer-weights (FATHMMcw) (*Ng & Henikoff, 2003*; *Shihab et al., 2013b*). Fourth, we focused on mutations affecting a set of 29 genes already shown to be implicated in GBM by a previous TCGA study (*Cancer Genome Atlas Research Network, 2008*). Mutations falling into coding regions of these 29 genes and showing high likelihood of altered protein function were assumed to be the best GBM-related mutations and potential cancer-drivers. Finally, we repeated this analysis on an independent validation dataset consisting of 15 pairs of RNA-seq and WES data from TCGA.

We generated paired RNA-seq and WES data from one GBM tumor (SD01) collected at St. David's Medical Center (Austin, TX, USA) after written informed consent, in a study approved by the Institutional Review Boards of St. David's Medical Center and of the University of Texas at Austin (approval numbers AMIRB 10-5-03 and 2012-01-0040). For WES and RNA-seq, we used the exome capture kit NimbleGen SeqCap EZ (Roche, Pleasanton, CA, USA) and the NEBNext small RNA kit (NEB, Ipswich, MA, USA), respectively. Sequencing was carried out at the NGS Core Facility of the MD Anderson Cancer Center Science Park on an Illumina HiSeq 2500. Data is available in dbGaP (https://www.ncbi.nlm.nih.gov/projects/gap/cgi-bin/study.cgi?study_id=phs001389.v1.p1). For GBM data from TCGA, BAM files resulting from alignment were downloaded from the Genomic Data Commons data portal and used directly in the subsequent analysis pipeline since they were already aligned with STAR. To evaluate variants, two databases were used: dbSNP (*Kitts et al., 2013*) with the b147 build on the GRCh38 reference ($37 \times 10^6$ variants), and the COSMIC database v78 (*Forbes et al., 2015*), which contains $3.3 \times 10^6$ known somatic variants. We carried out all analyses using the GRCh38 primary assembly reference acquired from GENCODE (*Harrow et al., 2012*). ANNOVAR (v.2016Feb01)

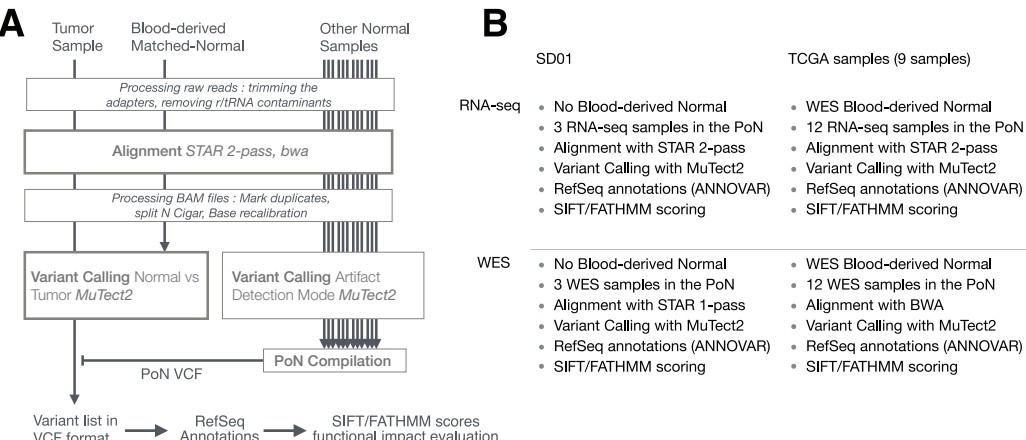

**Figure 1** **Pipeline used to detect RNA-seq variants.** (A) Principal steps in the pipeline used to identify and annotate somatic mutations. Mutation calling was done for each paired tumor sample/matched-normal. An RNA-seq-specific panel-of-normals (PoN) and a WES-specific PoN were generated. (B) Distinction between pipelines and their associated methodologies for SD01 and TCGA samples, and the difference between RNA-seq and WES pipeline used in this study.

(*Wang, Li & Hakonarson, 2010*) was used to annotate variants relative to RefSeq annotations (release 73) (*O'Leary et al., 2016*).

## A pipeline to detect variants from RNA-seq data with STAR 2-pass and GATK MuTect2 and distinguish GBM-related mutations

The general pipeline used is shown in Fig. 1, with slight differences between samples (SD01 and TCGA) or techniques (RNA-seq and WES) as depicted in Fig. 1B. The workflow was adapted from GATK best practices for variant calling (*Van der Auwera, 2014*; *Van der Auwera et al., 2013*) but using MuTect2 for variant calling. The process first involved trimming the adapters with cutadapt (v1.10) (*Martin, 2011*) from fastq files, removing sequences that were shorter than 36 bases after trimming, and removing rRNA and tRNA sequences by aligning with BWA (v0.7.12-r1039) (*Li & Durbin, 2009*) to a reference built with known rRNA/tRNA. Filtered reads were then aligned with STAR aligner (v2.4.2a) using a 2-pass procedure (*Dobin & Gingeras, 2015*). Before variant calling, aligned reads in BAM format were sorted, duplicate reads were flagged (MarkDuplicates, Picard v2.5.0), the base scores recalibrated (BaseRecalibrator, GATK v3.6) and RNA-seq reads were split into exons (SplitNCigarReads, GATK v3.6). Variant calling was done with MuTect2 in tumor versus normal mode as described below. Variants recovered in VCF files were then separated into RNA-seq-only, Intersection and WES-only. ANNOVAR (v.2016Feb01) (*Wang, Li & Hakonarson, 2010*) was used to annotate variants relative to RefSeq annotations (release 73) (*O'Leary et al., 2016*). SIFT score/prediction (v2.3) (*Ng & Henikoff, 2003*), and FATHMM score/prediction with cancer weights (v2.3) (*Shihab et al., 2013a*, *2013b*) were used to evaluate the functional impact of non-synonymous SNVs and frameshift indels. Finally, a set of 29 genes known to be related to GBM (*Cancer Genome Atlas Research Network, 2008*) was used to evaluate GBM-related mutations in specific pathways.

## Variant calling using MuTect2 from genome analysis toolkit

MuTect2 infers genotypes with two log-odd ratios (*Cibulskis et al., 2013*) which score the confidence that a mutation is present in the tumor sample (TLOD score) and is absent from the matched-normal sample (NLOD score). The thresholds used by MuTect2 to consider a variant as being real and somatic (leading to the annotation "PASS") are by default TLOD > 6.3 and NLOD > 2.2. For dbSNP variants, a higher NLOD threshold of 5.5 is used, except if the variant is also present in the COSMIC database.

## Building a panel of normals for variant calling with MuTect2

The creation of a Panel of Normals (PoN) is an optional step that improves variant calling by filtering out method-specific artefacts, by doing variant calling (MuTect2) on a set of normal samples (Fig. 1A). The samples for the PoN should ideally be obtained through protocols and data processing steps closely matched to the tumor sample. For this reason, two PoN were built, one with RNA-seq data from normal samples and another with WES data from normal samples. Then, variants identified by MuTect2 in at least two normal samples were compiled together into one PoN VCF file. Although using 30 normal samples is recommended by GATK, we used only 12 normal samples as they were matched to the 12 GBM tumor samples from TCGA.

## MuTect2 filters

Based on the TLOD score, MuTect2 will reject a variant when a specific TLOD > 6.3 threshold is not reached, suggesting insufficient evidence of its presence in the tumor sample (*t_lod_fstar* filter). *homologous_mapping_event* is a filter that detects homologous sequences and filters out variants falling into sequences that have three or more events observed in the tumor. *clustered_events* is a filter for clustered artifacts. *str_contraction* filters out variants from short tandem repeat regions. *alt_allele_in_normal* filters out variants if enough evidence is shown of its presence in the normal sample (NLOD threshold > 2.0). *multi_event_alt_allele_in_normal* filters out a variant when multiple events are detected at the same position in the matched-normal sample. *germline_risk* filters out variants that show sufficient evidence of being germline based on dbSNP, COSMIC and the matched-normal sample (NLOD value). *panel_of_normals* filters out variants present in at least two samples of the panel of normals.

## RefSeq annotations with ANNOVAR

ANNOVAR (v.2016Feb01) (*Wang, Li & Hakonarson, 2010*) was used to annotate the variants in the VCF file with RefSeq Genes annotations (release 73 with reference GRCh38) (*O'Leary et al., 2016*) and SIFT scores/predictions (v2.3) (*Ng & Henikoff, 2003*). RefSeq gives the closest gene name, or the two closest genes whenever a variant falls within intergenic regions. RefSeq also gives information about the type of mutation and the eventual amino acid change, whenever a variant falls in a coding region. For effects on alternative splicing, RefSeq gives a list of all possible transcripts.

## Scoring non-synonymous SNVs and indels with SIFT score

One way to assess the functional impact of an amino-acid (AA) change is to use SIFT (*Ng & Henikoff, 2003*), which uses homologous sequence comparison. SIFT (v2.3) gives a score based on the frequency at which an AA appears at a specific location in functionally related protein sequences. The AA change is given a predicted score: Tolerated ($p > 0.05$) or Deleterious ($p < 0.05$). Low scores typically occur in highly conserved regions that tend to be intolerant to most substitutions. On the contrary, unconserved regions tend to be more tolerant to AA changes. *SIFT indel* has been developed for scoring frameshifting indels (*Hu & Ng, 2013*), which relies on a different algorithm based on a machine learning model. It gives a prediction of damaging or neutral along with a confidence score.

## Scoring non-synonymous SNVs and indels with FATHMM cancer-weighted scores

Functional analysis through hidden markov models (FATHMM v2.3) also uses homologous protein sequences to find the probability of an amino acid substitution at a given position. The algorithm relies on Hidden Markov models to compute probabilities, its final scores being a ratio between the probability of the wild-type and the mutant AA. The version used here (*Shihab et al., 2013b*) also incorporates cancer weights (FATHMMcw), the frequency of cancer-associated variants from the CanProVar database and wild type weights, the frequency of neutral polymorphisms from UniRef database falling in the same protein region as the variants. The final score is an indication whether an AA substitution is deleterious and associated with cancer (prediction CANCER given for score $< -0.75$) or neutral (prediction PASSENGER given for score $> -0.75$). FATHMM for indels (*Shihab et al., 2015*) works on indels shorter than 20 bp and emits a prediction (pathogenic or neutral) together with a confidence score (expressed in %).

## Criterion to build a set of 29 genes previously shown to be altered in GBM

A set of 29 genes that were shown to be the most frequently mutated genes in GBM by a TCGA study on 91 GBM samples (*Cancer Genome Atlas Research Network, 2008*) was used to look for somatic mutations in GBM-related pathways. Genes selected to be part of the set were ARF, BRCA2, CBL, CDK4, CDKN2B, CDKN2C, EGFR, EP300, ERBB2, ERBB3, FGFR2, IRS1, MDM2, MDM4, MET, MSH6, NF1, P16, PDGFRB, PIK3C2B, PIK3C2G, PIK3CA, PIK3R1, PRKCZ, PTEN, RB1, SPRY2, TP53 and TSC2. These genes were shown to bear mutations in at least 2% of samples, the most altered being ARF (49%), EGFR (45%), PTEN (36%) and TP53 (35%). The "Best GBM-related mutation" (Table 1) is indicated when a mutation was included in this set of 29 genes, part of COSMIC database but not in dbSNP, resulted in an AA change and retained based on both SIFT and FATHMM scores as being functionally deleterious for protein function.

**Table 1** "Best GBM-related mutations" from coding regions of SD01 and TCGA samples.

| Gene | Sample | AA change | FATHMM score | SIFT score | AF (Tumor) | Coverage (Tumor) |
|------|--------|-----------|--------------|------------|------------|------------------|
| EGFR | SD01 RNA-seq only | A702S | −0.97 (CANCER) | 0.01 (Del) | 0.015 | 852 |
| EGFR | SD01 Intersection | A289V | −1.04 (CANCER) | 0.002 (Del) | 0.072 | 125 |
| EGFR | GBM01 Intersection | G63R | −1.93 (CANCER) | 0.0 (Del) | 0.175 | 296 |
| TP53 | GBM01 Intersection | G105R | −10.02 (CANCER) | 0.0 (Del) | 0.44 | 50 |
| TP53 | GBM02 RNA-seq only | I254S | −9.48 (CANCER) | 0.0 (Del) | 0.949 | 390 |
| TSC2 | GBM02 RNA-seq only | V296fs | 71% (pathogenic) | 85.8% (Dam) | 0.137 | 55 |
| PTEN | GBM02 Intersection | D107Y | −3.06 (CANCER) | 0.0 (Del) | 0.69 | 92 |
| PTEN | GBM03 Intersection | R173H | −6.42 (CANCER) | 0.0 (Del) | 0.331 | 173 |
| PTEN | GBM04 Intersection | D326fs | 88% (pathogenic) | 85.8% (Dam) | 0.393 | 146 |
| PTEN | GBM07 WXS only | R130Q | −5.84 (CANCER) | 0.0 (Del) | 0.713 | 190 |
| NF1 | GBM10 WXS only | C622F | −0.83 (CANCER) | 0.01 (Del) | 0.403 | 389 |

Notes:
All variants shown were included in COSMIC and in a set of 29 GBM-related genes but not dbSNP. All variants are deleterious based on scoring by SIFT and FATHMM with cancer weights. For SIFTindel and FATHMM indels, the score is given as a confidence score of the prediction. AF (Allele Fraction, tumor) shows the proportion of altered reads in tumor samples, with Coverage (tumor) being the total number of reads at the variant position.

# RESULTS

## Read counts and variant features highlight differences between RNA-seq and WES variants in TCGA samples

In the majority of samples, RNA-seq showed fewer uniquely mapped reads than WES (Fig. 2A; Fig. S1). Secondary alignments and unmapped reads were generally higher in the RNA-seq data, which could be due in part to unmapped splice junction reads and mismatches in RNA-seq due to RNA editing. Adenosine to inosine is the most common form of RNA editing in humans, leading mainly to A > G and T > C base substitutions (Picardi et al., 2015), which were clearly enriched in RNA-seq compared to WES data (Fig. 2B). RNA editing site databases like DARNED (Kiran et al., 2013), RADAR (Ramaswami & Li, 2014) or Inosinome Atlas (Picardi et al., 2015) could potentially be used to filter out such variants (Piskol, Ramaswami & Li, 2013).

The proportion of variants filtered by the different MuTect2 filters are shown in Fig. 2C. MuTect2 generates two log-odd ratios, TLOD and NLOD, which can be used to infer the somatic origin of a variant (Materials and Methods). RNA-seq variants showed lower TLOD scores and slightly higher NLOD scores than WES variants. Low read counts or poor base qualities supporting the altered allele in tumor can lead to low TLOD values. Fewer RNA-seq variants met the TLOD threshold (Fig. 2C, TLODfstar). Interestingly, TLOD scores of COSMIC variants were higher than non-COSMIC variants (Fig. S2), suggesting that TLOD reflects the higher true positive rate. On the other hand, variants that also occur in the matched-normal samples could be filtered by the AltAlleleInNormal MuTect2 filter based on NLOD values. RNA-seq data from TCGA samples showed particularly low numbers of variants excluded by this filter (Fig. 2C), which could be due to coverage differences between tumor RNA-seq and matched-normal (the latter being

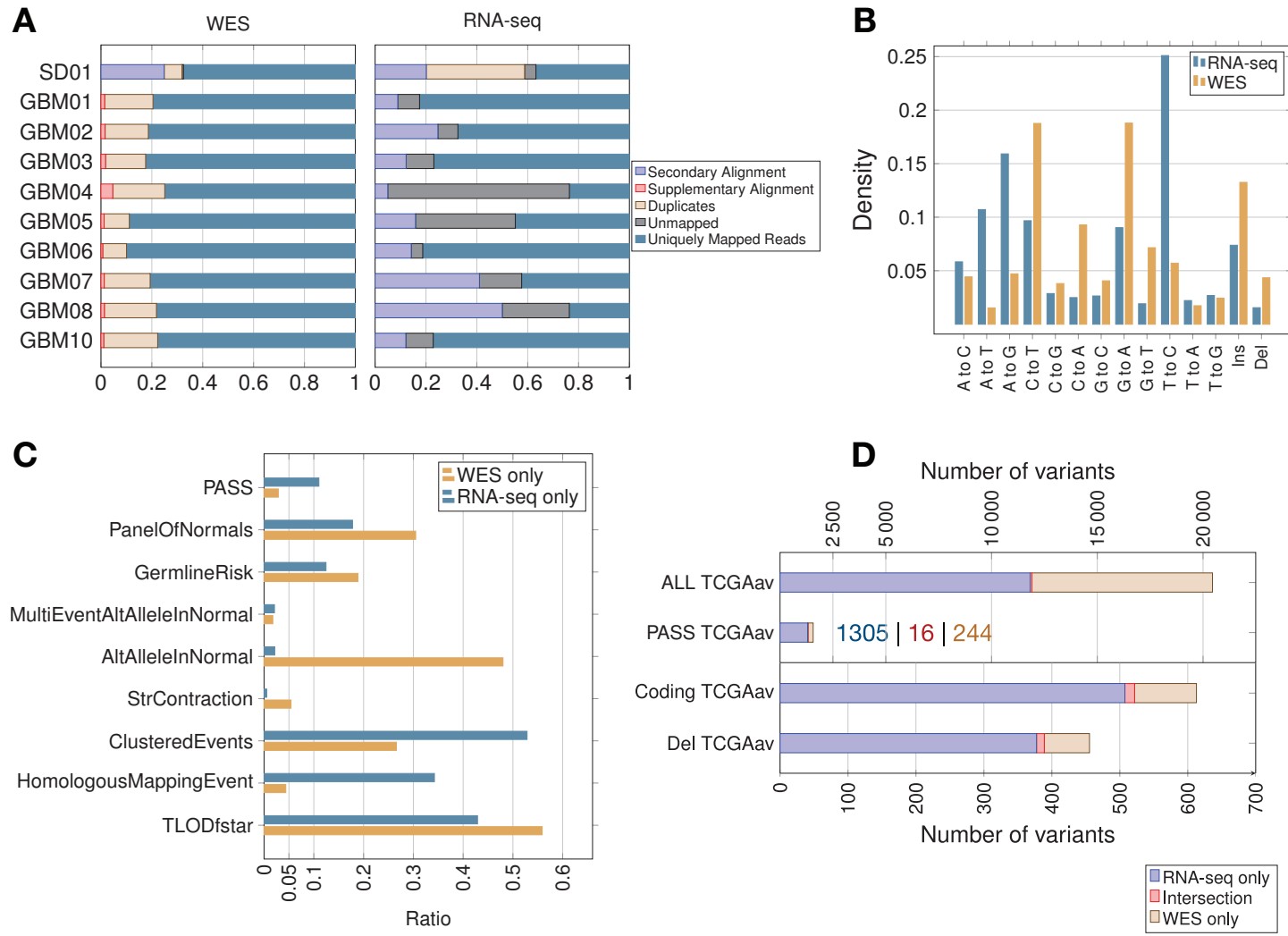

**Figure 2 Read count, filtering by MuTect2, mutation spectrum and total variant count in GBM samples.** (A) Proportion of reads (using samtools on BAM files) before variant calling. (B) Mutation spectrum indicating the type of base substitution in total RNA-seq and WES data. The *Y*-axis shows the proportion of mutations. (C) MuTect2 filtering statistics. Proportion of variants failing each MuTect2 filter. PASS stands for the proportion of variants accepted as true and somatic by MuTect2. The other filters are described in "Materials and Methods." (D) Total number of variants for TCGA samples (averaged over nine samples—TCGAav). ALL is the number of variants before MuTect2 filtering. PASS are the ones accepted as true and somatic by MuTect2. The number scale on top refers to the top two classes of variants (ALL and PASS). Coding refers to variants from coding regions. Del stands for variants in coding regions inducing an AA change (non-synonymous SNVs, frameshift indels or stop gain/loss). The number scale on bottom refers to the bottom two classes of variants (Coding and Del). The actual number of variants in PASS TCGAav from RNA-seq (blue), intersection (red) and WES (beige) is also indicated.     

WES data). Thus, distinct variant features given as an output by MuTect2 could be used to build a variant filtering model (*Ding et al., 2012*).

Variants accepted as true and somatic (PASS) by MuTect2 were higher in RNA-seq than WES for all TCGA samples (Fig. 2D). The overlap between RNA-seq and WES was small in all samples, but interestingly, the overlap increased with increasing significance of the variants. An average of only 6.60% of WES variants retained by MuTect2 (PASS) were also present in RNA-seq, while 15.9% of WES variants from coding regions and 17.2% of functional mutations were common to RNA-seq (Fig. 2D).

Coverage differences between RNA-seq and WES could partially explain the phenomenon. A previous study indeed found that ~71% of RNA-seq variants fell outside the WES capture boundaries (O'Brien et al., 2015). Moreover, they showed that a high proportion of RNA-seq-only variants were missed by WES because of their low allele fraction (AF).

As expected, the RNA-seq/WES intersection was enriched in variants from coding regions (89.7% of coding variants), since both RNA-seq and WES query exons. RNA-seq data also showed an unexpected level of intronic/intergenic variants. Intronic mRNA reads could partly come from unspliced RNA (pre-mRNA). A previous study has indeed detected many intronic mRNA variants, which could come from inefficient splicing in cancer (Sowalsky et al., 2015). On the other hand, intergenic RNA-seq variants could come from unannotated genes, non-coding RNA, retrotransposons, splicing errors (Pickrell et al., 2010) and sequencing/mapping errors.

## Allele fraction and coverage are useful features to further classify variants

In theory, heterozygous mutations would show an AF around 0.5. However, somatic mutations from cancer cells are expected to appear at lower frequencies, as tumor samples are heterogeneous and not pure clones. Moreover, CNVs can lead to gain/loss of chromosomes and/or duplications of genes (Yin et al., 2009). RNA-seq-only variants showed a surprising AF distribution in that 38.2% showed an AF > 0.95 (Figs. 3A and 3B) versus only 0.50% of WES-only variants. These high AF RNA-seq-only variants showed low coverage, and the majority of them occurred in intronic/intergenic regions (85.6% of RNA-seq-only variants with AF > 0.95). Conversely, we also found a high number of RNA-seq-only variants showing AF < 0.05 (36.3% of RNA-seq variants representing 4,267 variants in nine TCGA samples). In comparison, WES-only data showed only 502 variants (22.8%) with AF < 0.05. Low AF RNA-seq-only variants mainly originated from coding regions (81.1% of RNA-seq-only variants with AF < 0.05) and often showed high coverage, which distinguished them from WES-only variants (Fig. 3A). We examined the coverage data for RNA-seq-only variants with AF < 0.05 and coverage > 500, and found only one variant (out of 2,192) that was also present in WES data, and showing only one altered read. This region of high coverage/low AF is of particular interest as it is likely to contain true somatic mutations that are missed in WES data.

## COSMIC/dbSNP overlap can be used as an indicator of the somatic/germline content in TCGA samples

For each of the three classes of variants—WES-only, Intersection and RNA-seq-only—we examined the proportion in different genomic regions (Fig. 3C), potential for affecting protein function (Fig. 3D) and representation in dbSNP and COSMIC databases (Fig. 3E). The proportion of variants included in the dbSNP database is potentially an indicator of germline content among identified variants, while the overlap with the COSMIC database can serve as an indicator of somatic mutations (Fig. 3E). It must be noted that with the increasing coverage in dbSNP of variants from ever-increasing numbers of human genomes, inclusion in dbSNP cannot always rule out a somatic variant

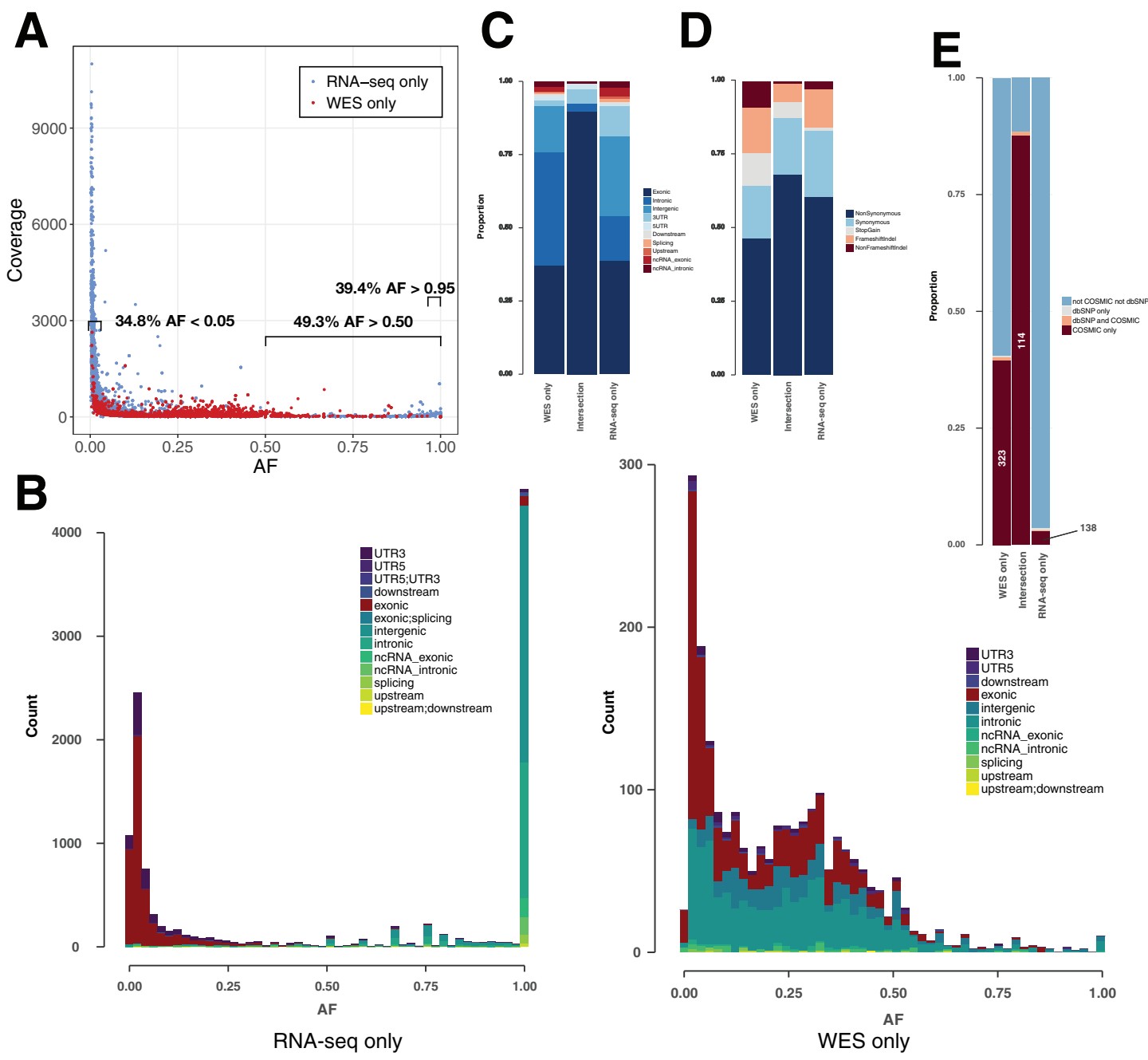

**Figure 3** **Variant features including allele fraction, coverage, genomic location and COSMIC/dbSNP content in TCGA samples.** (A) Scatter plot representing the fraction of the altered allele estimated from altered read fraction (allele fraction) versus coverage at the variant position (total number of reads). The indicated intervals show the proportion of RNA-seq only variants having AF < 0.05, AF > 0.50 or AF > 0.95. Merged data from nine TCGA samples is shown. (B) Histograms of the distribution of allele fraction (AF) for the indicated classes of variants. Merged data from nine TCGA samples is shown. (C) Genomic location of PASS variants, given as the average value over nine TCGA samples. (D) Type of variants from coding regions, given as the average obtained over nine TCGA samples. Indel stands for insertions and deletions. (E) Proportion of variants from coding regions included in COSMIC and/or dbSNP, given as the average obtained over nine TCGA samples. Absolute numbers of COSMIC-only variants are indicated.

(*Nadarajah et al., 2016*). Nevertheless, their overlap is small, at least in the versions of the databases we used, with only 0.15% of dbSNP variants included in COSMIC (Fig. S3). Coding variants identified by both RNA-seq & WES (Intersection) showed a particularly high proportion (87.7%) included in COSMIC but not in dbSNP (COSMIC-only), which may be considered the most likely candidates for somatic mutations. A high proportion of WES-only coding variants (39.5%) and a low proportion of RNA-seq-only coding variants (3.0%) were likewise found in COSMIC-only but although the proportions were very different, both WES-only and RNA-seq-only variants contained the same order-of-magnitude COSMIC-only variants (Fig. 3E). Thus, RNA-seq-only identified 138 COSMIC-only variants from coding regions that were therefore missed by WES-only. Because COSMIC contains variants discovered mainly by WES, it is possible that many of the RNA-seq-only variants unknown to both COSMIC and dbSNP, representing 96.4% of RNA-seq-only variants from coding regions (4,402 variants in nine samples), could include many bonafide cancer somatic mutations. We therefore explored this possibility further.

## Genes showing somatic mutations in multiple TCGA samples only in RNA-seq data

There were 63 genes with RNA-seq-only variants that were mutated in five or more tumors, and many genes from this group have been implicated in cancer (Fig. 4). For example, a set of three complement related genes—complement C3, α-2 macroglobulin and the complement lysis inhibitor SP-40/clusterin (CLU)—that have been implicated in various cancers including gliomas (*Reis et al., 2018*; *Saratsis et al., 2014*; *Shinoura et al., 1994*; *Suman et al., 2016*) were present in this group, and interestingly, these three proteins have been recently shown to form a network of related biomarkers in B-ALL (*Cavalcante et al., 2016*). One tumor contained a cluster of highly mutated genes (Fig. 4, bottom left), including SPARC and FLNA, which are associated with cell-matrix interactions and cell motility (*Neuzillet et al., 2013*; *Xu et al., 2010*), and thus possibly involved in metastasis. On the other hand, MAGED1 was linked with cell-death mechanisms (*Mouri et al., 2013*), which are often disrupted in cancer. One frameshift insertion was detected in the ARF1 gene located at the exact same position (G14fs) in all nine samples. This was a COSMIC-only variant with plausible AF and coverage. Despite high coverages in WES at the variant position, the insertion was never present in WES data, and since indels have been shown to be more prone to artefacts (*Kroigard et al., 2016*), it was not retained in Tables 1 and 2 (see below). Note that the mutational landscape presented here is distinct from the one obtained by a TCGA study on WES data (*Brennan et al., 2013*), which is not surprising as RNA-seq-only data is likely interrogating other regions of the genome relative to WES.

## Analysis of somatic mutations found by RNA-seq without a corresponding matched normal sample

The SD01 GBM tumor sample had no corresponding matched normal to enable reliable distinction of somatic mutations from germline variants, so it presented unusual

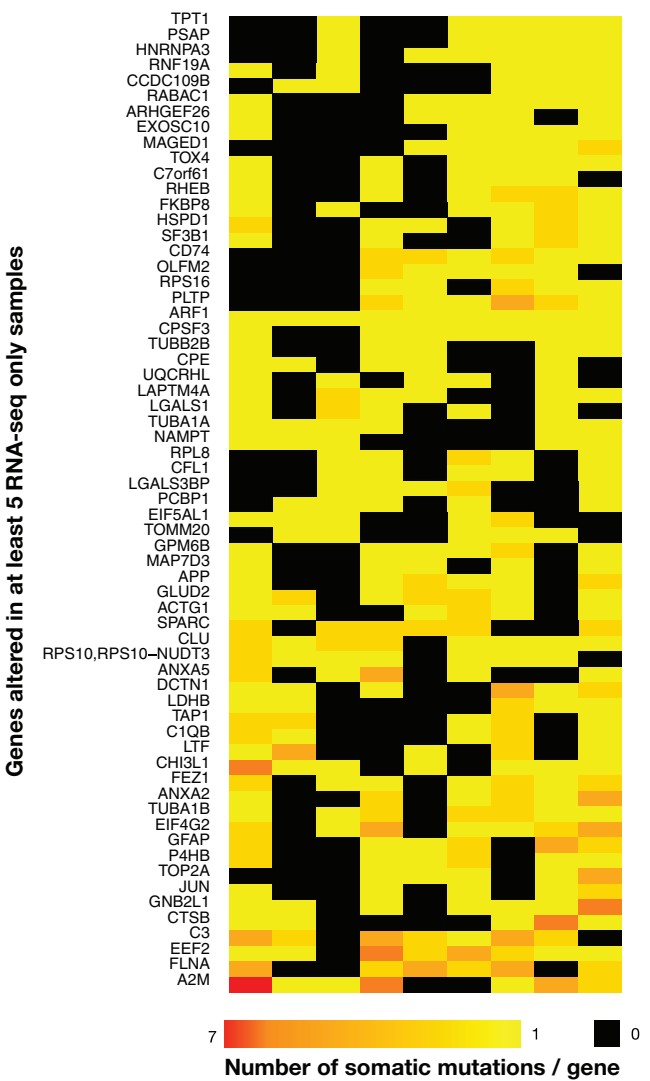

**Figure 4 Heatmap of the 63 most widely mutated genes across TCGA samples in RNA-seq-only.** Only genes altered in coding regions in at least five out of nine tumor samples are shown. Rows indicate genes and columns are tumors. Black indicates no variants, bright yellow only one variant/gene and red seven variants/gene, with a gradient from yellow to red indicating a number of variants/gene included between one and seven. The heatmap was clustered by rows and columns (the dendrogram is not shown).

challenges. However, it is worthwhile to consider such samples because often, RNA-seq data may be available from a tumor without a corresponding matched normal sample. The total number of variant called in SD01 was much higher than the average TCGA sample (by 10.5-fold for RNA-seq and 17.7-fold for WES). SD01 had a similar number of aligned reads as the TCGA samples for both RNA-seq and WES, so the higher number of somatic variants could be in part due to the absence of matched-normal, the small panel of normals used and/or by a higher underlying mutation rate in this specific tumor. MuTect2 variant calling was carried out in tumor-only mode and only relied on TLOD values without distinction between somatic and germline variants. Many dbSNP variants

**Table 2 Variants unknown by both COSMIC and dbSNP and candidates to be new GBM-related functional somatic mutations.**

| Gene | Sample | AA change | FATHMM score | SIFT score | COSMIC | AF (Tumor) | Coverage (Tumor) |
|---|---|---|---|---|---|---|---|
| EGFR | SD01 RNA-seq-only | S229fs | 93% (pathogenic) | 85.8% (Dam) | No (S229C) | 0.045 | 169 |
| EGFR | SD01 RNA-seq-only | W477fs | 51% (neutral) | 85.8% (Dam) | No (W477*) | 0.046 | 447 |
| PIK3C2 | SD01 WES-only | I255N | −3.49 (CANCER) | 0 (Del) | No | 0.433 | 64 |
| CDKN2C | GBM02 RNA-seq-only | V130A | −0.21 (PASSENGER) | 0.03 (Del) | No | 0.027 | 470 |
| PDGFRB | GBM02 RNA-seq-only | V840A | −2.34 (CANCER) | 0.23 (Tol) | No | 0.021 | 262 |
| RB1 | GBM03 RNA-seq-only | L872fs | 77% (pathogenic) | 85.8% (Dam) | No | 0.035 | 355 |
| EGFR | GBM05 RNA-seq-only | M600T | −1.69 (CANCER) | 0.38 (Tol) | No (M600V) | 8.1E-03 | 6,240 |
| EGFR | GBM05 RNA-seq-only | L718R | −2.85 (CANCER) | 0 (Del) | No (L718M) | 4.5E-03 | 4,792 |
| PDGFRB | GBM06 RNA-seq-only | Q1075R | −1.25 (CANCER) | 0.52 (Tol) | No | 0.058 | 90 |

Notes:
Variants included in the set of 29 GBM-related genes and not included in COSMIC or dbSNP are shown, although COSMIC contained alternative variants at the same positions for four mutations that were found by RNA-seq-only. For SIFTindel and FATHMM indels, the score is given as a confidence score. AF (tumor) shows the proportion of altered reads in tumor samples, with Coverage (tumor) being the total number of reads at the variant position. Allele Fraction and Coverage was used to further exclude potential artifacts, which are not listed here.

\* Indicates a nonsense mutation.

were indeed observed (Fig. S4). The distribution of SD01 variants by chromosome showed a remarkably high number of variants on Chromosome 7 (Fig. S5), which could reflect amplification of Chromosome 7, a common feature in GBM (*Cancer Genome Atlas Research Network, 2008*). SD01 also showed a higher density of transition variants (T > C, C > T, A > G and G > A), which tend to be less deleterious, as expected for germline variants (*Campbell & Eichler, 2013*). Nevertheless, SD01 RNA-seq-only variants included several interesting candidate somatic mutations. One of these RNA-seq-only mutations was EGFR-A702S, found in COSMIC but not in dbSNP, and retained by both SIFT and FATHMM scores (see below). Two other frameshift insertions were also found by RNA-seq-only data in EGFR (S229fs and W477fs), with COSMIC variants found at the same AA coordinates (Table 2). Moreover, the intersection between RNA-seq and WES data in SD01 showed other interesting candidates, such as a point mutation in EGFR (A289V—retained by both SIFT/FATHMM, and present in COSMIC but not in dbSNP).

## Analyzing the functional impact of somatic mutations on protein function in relation to cancer and GBM pathways

We used the algorithms FATHMM and SIFT to evaluate the potential impact of somatic variants on protein function in cancer pathways (Materials and Methods). The FATHMM and SIFT score distributions showed a significant difference only for FATHMM scores between the Intersection and WES-only (Fig. 5). Many RNA-seq-only variants scored below both FATHMM and SIFT thresholds, indicating they could be potential functional mutations. The overall proportion of variants retained by FATHMM and SIFT was higher for Intersection variants (11.8%, Fig. 5D), and slightly higher in RNA-seq-only than WES-only. Mutations present among a set of 29 hand-curated GBM-related genes were designated as the "best GBM-related mutations" (Table 1), and comprised 11 mutations. RNA-seq-only detected three of these 11 mutations, while WES-only found two and the Intersection between RNA-seq and WES found six of the 11 GBM mutations.
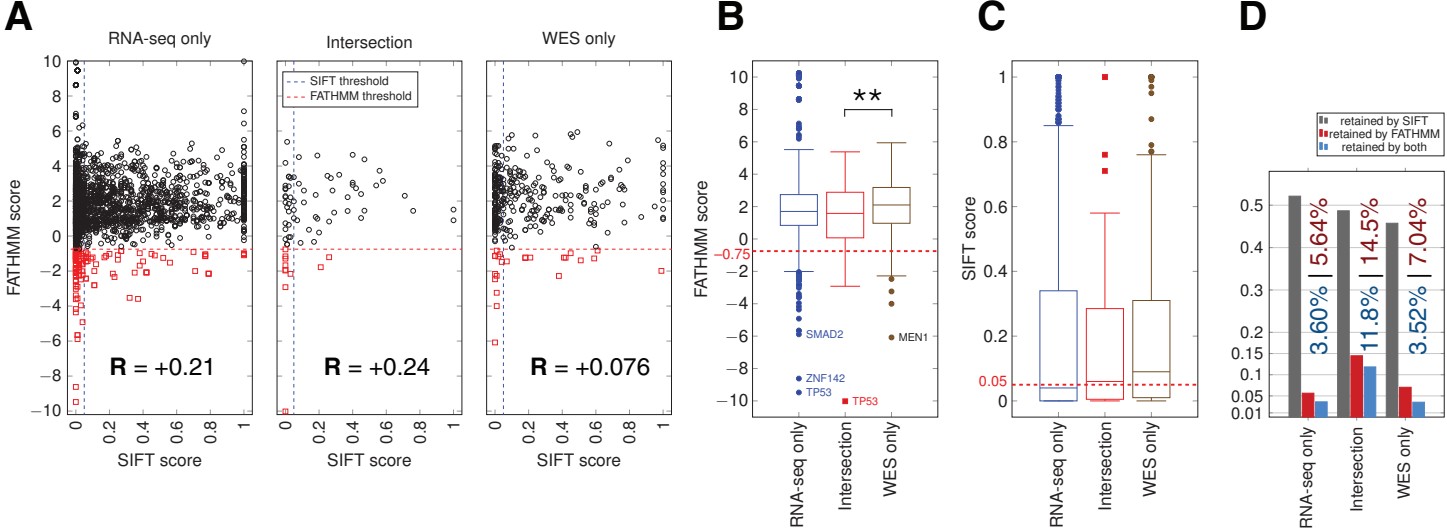

**Figure 5 SIFT and FATHMM scores for TCGA samples.** (A) Scatterplots of FATHMM scores versus SIFT scores for non-synonymous SNVs. The threshold for deleterious variants for SIFT is 0.05 and the threshold for cancer-drivers for FATHMM is −0.75, which are indicated. Mutations retained by both are thus in the bottom left corner of each plot. Pearson correlations between SIFT and FATHMM are indicated by *R* values for each plot. (B) Box plots showing the distribution of FATHMM scores for each of the indicated categories of variants. The only significant difference was between the Intersection and WES-only groups (*p* = 0.0085). Genes with the smallest variant scores are indicated. (C) Box plots showing the distribution of SIFT scores. No significant differences were observed. (D) Proportion of variants retained as deleterious or cancer-driver by SIFT and/or FATHMM respectively. The proportion retained by FATHMM is indicated in red, and the proportion retained by both SIFT and FATHMM is indicated in blue.

These three RNA-seq-only mutations (EGFR-A702S, TP53-I254S and TSC2-V296fs) are thus cancer-driver candidates found only by RNA-seq and should therefore motivate the use of RNA-seq as they were missed by WES. Taken together, our results suggest that the intersection between RNA-seq data and WES yielded the highest quality GBM-related mutations in TCGA samples, for three reasons. First, variants from RNA-seq/WES intersection showed 90.5% of COSMIC-only variants (Fig. 3E), an average much higher than WES-only or RNA-seq-only data. Second, coding variants from the intersection also showed more evidence of functional alteration through their SIFT and FATHMM scores (Fig. 5). Third, 6/11 of the "best GBM-related mutations" were identified in the intersection (Table 1), even though it was the smallest group in term of variant number. Thus combining RNA-seq and WES greatly improves the confidence in certain variants discovered by WES, particularly in highly expressed genes.

## New somatic/GBM-related mutations evaluation from unknown variants

Another group of findings are shown in Table 2 as being potentially undiscovered variants, as they were neither in COSMIC nor in dbSNP, but affected one of the 29 GBM-related genes and retained either by SIFT or FATHMM scores. These mutations are therefore the best candidates for being new discoveries as they implicate known GBM-related pathways. RNA-seq-only data allowed the discovery of 8/9 potentially new mutations, against only one new variant in WES-only data, which suggests that variant calling from
RNA-seq has considerable potential to generate new discoveries, including in already well known pathways. For example, an RNA-seq-only variant, EGFR-L718R, showed 22 variant reads out of a total of 4,792 (AF $4.5 \times 10^{-8}$). WES showed 101 reads at the same position, giving a probability of only 0.39 of at least one variant read occurring in the WES data (based on binomial probability). Interestingly, COSMIC has cataloged a different variant, L718M at the same position (Table 2).

In order to confirm the overall findings from the preceding analysis, we repeated the entire pipeline on an independent validation dataset comprising 15 GBM tumors downloaded from TCGA. The overall characteristics of the variants in this validation dataset, as well as the genomic locations, the nature of variants found in the WES, RNA-seq-only and Intersection sets, and the identities of genes showing significant RNA-seq-only variants matched well with our previous analysis (Fig. S6).

## DISCUSSION

Although WES has been the mainstay of somatic mutation identification in cancer genomes, our study suggests that variant calling from RNA-seq offers a valuable complement. RNA-seq revealed new variants that were clearly associated with GBM biology, were found at the same positions as previously known variants, and yet were missed by WES. A major reason for the ability of RNA-seq to identify new somatic variants likely comes from the higher sequencing coverage of strongly expressed genes. Oncogenes in cancers, such as EGFR in GBM, are likely to be highly expressed, and RNA-seq naturally provides better coverage of such genes than WES, and hence higher statistical confidence to detect variants. Additionally, even when tumor cells expressing active oncogenes comprise only a subset of the tumor, RNA-seq reads can capture this overrepresentation when RNA is isolated from the bulk tumor, whereas DNA used for WES cannot. In this regard, RNA-seq is likely to be advantageous even over whole-genome sequencing, where it is harder to achieve the same depth of coverage over all genes as WES.

The RNA-seq variants we identified in our analysis did not appear to have significantly lower quality than WES variants, although we saw a high number of variants with AF > 0.95 and low coverage in RNA-seq data. Based on MuTect2 output, RNA-seq detected more somatic mutations than WES in the TCGA samples. However, some RNA-seq variants could be considered questionable, since RNA-seq data has been shown to be more prone to false positive calls (*Cirulli et al., 2010*), in part due to errors during the RNA to cDNA conversion, mapping mismatches, or RNA editing processes (*Danecek et al., 2012*). Indels are also a source of possible artefacts (*Kroigard et al., 2016*) even though the local de novo assembly done by MuTect2 should reduce this artefact. Comparison of variants with known somatic mutations from the COSMIC database showed that WES-only data contained more COSMIC variants than RNA-seq-only in TCGA samples (323 versus 138; Fig. 3E). However, this representation is likely to be skewed by the fact that COSMIC variants were primarily discovered by WES. Variants in coding regions were represented in the same proportions in RNA-seq and WES (see Fig. 3C) and overrepresented in the intersection, suggesting that RNA-seq and WES coverage have a higher overlap in coding regions, and making it possible to compare mutations found in

both datasets within coding regions. We focused on variants causing an AA change, for which functional impact could be estimated with the scoring systems SIFT and FATHMM. To assess the identification of potential cancer-drivers that were specific to GBM, we evaluated the recovery of variants in 29 genes within specific pathways previously shown to be altered in GBM by a TCGA study (*Cancer Genome Atlas Research Network, 2008*). By this measure, RNA-seq-only data detected three out of 11 possible variants while WES-only detected two out of 11, even though COSMIC variants have been primarily discovered through WES. The intersection recovered six out of these 11 variants (see Table 1). Strikingly, RNA-seq-only data outperformed WES-only in discovering new mutations falling into these 29 GBM-related genes (8/9 findings). RNA-seq-only is thus able to not only detect already known mutations, but detect potentially new mutations falling into known GBM-related pathways, despite the high sequencing depth of WES. In sum RNA-seq was able to find nine of 11 key known mutations and eight new discoveries, justifying its use for variant discovery in cancer. RNA-seq data had the potential to better detect variants showing very low allelic fraction (*Cirulli et al., 2010*), when more reads were available in highly expressed genes. Analysis on the coverage indeed showed numerous variants showing low AF and high coverage and therefore likely to be missed by WES alone. Additionally, a previous study has shown that RNA-seq-only variants tend to be missed by WES mainly because they fall outside WES capture kit boundaries (~71% of RNA-seq-only variants versus WES), and tend to be located in highly expressed genes, which are more likely to be related to cancer than unexpressed genes, the ones falling into WES-only data (*Cirulli et al., 2010*).

Several ways of improving the detection of cancer-related mutations using RNA-seq are possible. First, it may be possible to optimize the pipeline by reducing artefacts and germline content. A recent study developed a pipeline for analysis of variants in RNA-seq data (*Piskol, Ramaswami & Li, 2013*). They used an indel realignment step and called variants in a more permissive way for RNA-seq but at the same time requiring better base quality scores. After variant calling, they filtered out known RNA editing sites using the RADAR database (*Ramaswami & Li, 2014*). Second, a variant filtering step using a machine-learning approach could be used to train a model with MuTect2 output features specifically for RNA-seq data (*Spinella et al., 2016*). Third, RNA-seq read generators such as BEERS (*Grant et al., 2011*) or Flux simulator (*Griebel et al., 2012*) could be used to optimize the pipeline by fine-tuning the sensitivity/specificity.

## CONCLUSIONS

Somatic mutations in tumors can be identified from RNA-seq data as a complement to exome sequencing. Some mutations we identified in GBM based on RNA-seq data occurred in genes known to be related to GBM and were missed by exome sequencing alone. In many cases, different variants at the same positions were cataloged in the COSMIC database of somatic mutations in cancer. The use of RNA-seq can thus potentially reveal new somatic mutations underlying cancer. Our work suggests that since the majority of studies on cancer-driving mutations used WES-only, they are likely to have

missed some key driver mutations that might be found using complementary RNA-seq datasets from the same tumors.

## ACKNOWLEDGEMENTS

We thank M. Shpak, M. Cowperthwaite and A.W. Hall for tumor specimens and data. We thank the Next Generation Sequencing Core Facility at the University of Texas MD Anderson Cancer Center Science Park for sequencing, and the Texas Advanced Computing Center (TACC) and the Biomedical Research Computing Facility (BRCF) at The University of Texas at Austin for HPC resources and computational support. This work is based in part based upon data generated by The Cancer Genome Atlas managed by the NCI and NHGRI for which we are grateful.

### Funding

This work was funded in part by grants from the National Institutes of Health (CA198648) and the Cancer Prevention Research Institute of Texas (RP120194) to Vishwanath R. Iyer. The funders had no role in study design, data collection and analysis, decision to publish, or preparation of the manuscript.

### Grant Disclosures

The following grant information was disclosed by the authors:
National Institutes of Health: CA198648.
Cancer Prevention Research Institute of Texas: RP120194.

### Competing Interests

The authors declare that they have no competing interests.

### Author Contributions

- Alexandre Coudray conceived and designed the experiments, performed the experiments, analyzed the data, prepared figures and/or tables, authored or reviewed drafts of the paper, approved the final draft.
- Anna M. Battenhouse contributed reagents/materials/analysis tools.
- Philipp Bucher contributed reagents/materials/analysis tools.
- Vishwanath R. Iyer conceived and designed the experiments, authored or reviewed drafts of the paper, approved the final draft.

### Human Ethics

The following information was supplied relating to ethical approvals (i.e., approving body and any reference numbers):

This study was approved by the Institutional Review Boards of St. David's Medical Center and of the University of Texas at Austin (approval numbers AMIRB 10-5-03 and 2012-01-0040).

## DNA Deposition

The following information was supplied regarding the deposition of DNA sequences:

The aligned bam files and raw fastq files from the GBM dataset generated and analyzed in this study are available in the dbGaP repository: https://www.ncbi.nlm.nih.gov/projects/gap/cgi-bin/study.cgi?study_id=phs001389.v1.p1.

## Data Availability

The research in this article did not generate any data or code. We used existing software as described in the Materials and Methods.

## Supplemental Information

Supplemental information for this article can be found online at http://dx.doi.org/10.7717/peerj.5362#supplemental-information.

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
