# Peer review of "Detection and benchmarking of somatic mutations in cancer genomes using RNA-seq data"

_PeerJ, doi:10.7717/peerj.5362_

## Round 0.1 · original submission · Major Revisions

In particular, two external reviewers carefully evaluated your work and highlight some key points that deserve attention, such as i) a critical discussion and analyses when it comes to the usage of dbSNP as a source of germline mutations, ii) the need for a validation data set, iii) a careful revision of the text, references and figures, iv) comparison with other variant callers, v) to deposit (and provide the link) the internal dataset in an open access repository.

Reviewer 1 ·

Basic reporting

Clear and professional English language used throughout the article. Literature references are not sufficient in the paper. For example, line 44: you referred to several studies without references. On line 53 provide the reference for MuTect2 (Cibulskis
et al., Nature Biotechnology, 2013) although you provided it on line 116.Please add references when needed.
Figure 3 (A and B) is really small and it is difficult to see the different variant features. Please improve it.

Experimental design

Primary research is within scope of PeerJ as it is a research article. The paper presents a pipeline for processing RNA-seq data from glioblastoma multiforme tumors in order to identify somatic mutations and entails the use of the STAR aligner 2-pass procedure jointly with MuTect2 from GATK to detect somatic variants.They compared the performance of their pipeline to whole-exome sequencing.

Validity of the findings

RNA seq can give similar results than WES under certain conditions and this article provides another example for it.
In the results section, read counts and variant features highlight differences between RNA-seq and WES variants in TCGA samples, the author wrote line 218: “The RNA-seq/WES intersection was enriched in variants from coding regions (89.7% of coding variants), which was probably induced by a higher coverage overlap”. As RNA seq captures RNA, introns, exons, the RNA-seq/WES intersection is located in the CDS.

In the paragraph called “allele fraction and coverage are useful features to further classify variants” (line 225-239), did the authors checked whether reads supporting some of the high coverage low AF variants found with RNA-seq / not with WES could be found in the alignment files and did the authors try alternative variant callers that are highly sensitive at low AF frequencies, e.g LoFreq?

Wilm et al. LoFreq. (2012). A sequence-quality aware, ultra-sensitive variant caller for uncovering cell-population heterogeneity from high-throughput sequencing datasets. Nucleic Acids Res.; 40(22):11189-201.

Additional comments

The paper entitled “detection and benchmarking of somatic mutations in cancer genomes using RNA-seq data” develops and evaluates a pipeline for processing RNA-seq data from glioblastoma multiforme tumors in order to identify somatic mutations compared to whole-exome sequencing. The pipeline entails the use of the STAR aligner 2-pass procedure jointly with MuTect2 from GATK to detect somatic variants. The overall level of the manuscript is good: even if it is quite simple, it is well written and some important considerations are highlighted. In the following, there is a list of points that the authors should answer:
(1) did the authors checked whether reads supporting some of the high coverage low AF variants found with RNA-seq / not with WES could be found in the alignment files and did the authors try alternative variant callers that are highly sensitive at low AF frequencies, e.g LoFreq?
(2) add references in the paper
(3) do the figure 3 to improve its size and readability

Reviewer 2 ·

Basic reporting

The paper is well written and the hypotheses of this work are well defined. However, in my opinion the introduction should contain the background knowledge behind this work. From line 57 to 77, the authors explain the method but this is not the methods section.

Experimental design

This work fits very well the aims of the journal and it's potentially very interesting and useful for the community. The experimental design is well described however, several points need to be clarified.

1) First of all, in lines 63-67 the authors reported "First, we estimated the proportion of germline/somatic mutations by comparison of identified variants to the dbSNP database (Kitts et al. 2013) which catalogs known germline variants, and the Catalogue Of Somatic Mutations in Cancer or COSMIC database (Forbes et al. 2015)". These two database were used to differentiate between germline and somatic mutations. The key point here is that based on the dbSNP/NCBI handbook, dbSNP includes both germline and somatic mutations: "dbSNP accepts submissions for all classes of simple sequence variation, and provides access to variations of germline or somatic origin that are clinically significant"
https://www.ncbi.nlm.nih.gov/books/NBK174586/

Since this is a crucial step in the pipeline, the authors should comment on that. If the percentage of somatic mutations in dbSNP is low, the authors should show that and evaluate how this bias can affect the rest of the analysis. All the downstream analyses could be affected by this crucial step.

2) The number of samples is very low. You need a validation set to verify your discoveries.

3) The authors don't mention if the data are deposited in some repositories already or not (the internal dataset).

4) In line 129 and 130 the authors don't motivate why they used GATK, since 30 normal samples is recommended by GATK, and you only used 12 normal samples.

5) Related to my comment 1, the statement in line 392 can be taken into account at the beginning of the pipeline by considering only the germline mutations in dbSNP.

Validity of the findings

The findings depend on the answer to my previous comments, especially comments 1 and 2.

Additional comments

In general, this pipeline is well described, implemented and potentially useful in order to use RNA-seq to identify somatic mutations. However, the authors have to address several points (see comments).

---

## Round 0.2 · accepted · Accept

The reviewers and I are satisfied with the new version of the article, which nicely addresses all the previous concerns.

# Reviewer 1 ·

Basic reporting

The authors have addressed all the issues I pointed out in the first review.

Experimental design

The authors have addressed all the issues I pointed out in the first review.

Validity of the findings

The authors have addressed all the issues I pointed out in the first review.

Additional comments

The authors have addressed all the issues I pointed out in the first review. Therefore the article meets the PeerJ criteria and should be accepted as is.

Reviewer 2 ·

Basic reporting

The authors have answered to all my comments.

Experimental design

The authors have answered to all my comments.

Validity of the findings

The authors have answered to all my comments.

Additional comments

All my comments/questions have been answered. I'm satisfied with the revision and therefore I suggest the publication of the work in its current version.